# Towards Out-of-Distribution Sequential Event Prediction: A Causal Treatment

**Chenxiao Yang**[1], **Qitian Wu**[1], **Qingsong Wen**[2], **Zhiqiang Zhou**[2], **Liang Sun**[2], **Junchi Yan**[1]*

[1]Department of Computer Science and Engineering, Shanghai Jiao Tong University
[2]DAMO Academy, Alibaba Group
`{chr26195,echo740,yanjunchi}@sjtu.edu.cn`,
`{qingsong.wen,zhouzhiqiang.zzq,liang.sun}@alibaba-inc.com`

## Abstract

The goal of sequential event prediction is to estimate the next event based on a sequence of historical events, with applications to sequential recommendation, user behavior analysis and clinical treatment. In practice, the next-event prediction models are trained with sequential data collected at one time and need to generalize to newly arrived sequences in remote future, which requires models to handle temporal distribution shift from training to testing. In this paper, we first take a data-generating perspective to reveal a negative result that existing approaches with maximum likelihood estimation would fail for distribution shift due to the latent context confounder, i.e., the common cause for the historical events and the next event. Then we devise a new learning objective based on backdoor adjustment and further harness variational inference to make it tractable for sequence learning problems. On top of that, we propose a framework with hierarchical branching structures for learning context-specific representations. Comprehensive experiments on diverse tasks (e.g., sequential recommendation) demonstrate the effectiveness, applicability and scalability of our method with various off-the-shelf models as backbones.

## 1 Introduction

Real-world problem scenarios are flooded with sequential event data consisting of chronologically arrived events which reflect certain behaviors, activities or responses. A typical example is user activity prediction [56; 12; 62; 20; 44; 64] that aims to harness user's recent activities to estimate future ones, which could help downstream target advertisement in online platforms (e.g., e-commerce or social networks). Predicting future events also plays a central role in some real situations such as clinical treatment [3] for promoting social welfare.

A common nature of (sequential) event prediction lies in the different time intervals within which training and testing data are generated. Namely, models trained with data collected at one time are supposed to predict next event in the future [59; 36; 41] where the underlying data-generating distributions may have gone through variation due to environmental changes. However, most existing approaches [20; 14; 37; 24; 44; 39; 19; 18] overlook this issue in both problem formulation and empirical evaluation, which may leave the real problems under-resolved with model mis-specification, and result in over-estimation for model performance on real data.

As a concrete example in sequential recommenders [11], the data-generating distribution for user clicking behaviors over items is highly dependent on user preferences that are normally correlated

---

*The SJTU authors are also with MoE Key Lab of Artificial Intelligence, SJTU. Junchi Yan is the correspondence author who is also with Shanghai AI Laboratory.

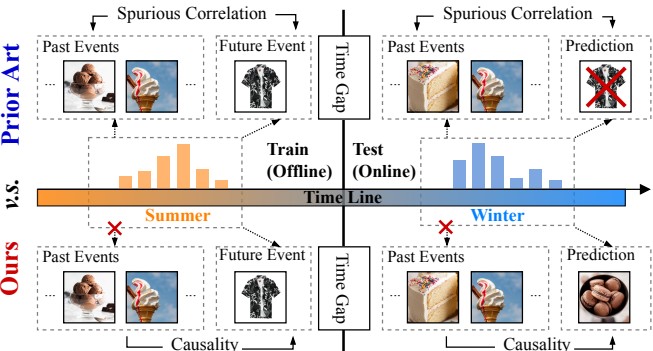

Figure 1: A toy example in recommendation. Prior art would spuriously correlate non-causal items ('ice cream' and 'beach T-shirt') and produce undesired results under a new environment in the future. Our approach endeavors to alleviate the issue via counteracting the effects of contexts (seasons).

with some external factors, like the dynamic fashion trends in different years [13] or seasonal influences on item popularity [43]. These highly time-sensitive external factors would induce distinct user preferences leading to different behavioral data as time goes by. This issue could also partially explain the common phenomenon of model performance drop after adaptation from offline to online environments [41; 36; 22].

Handling distribution shift in sequential event data poses several non-trivial challenges. *First*, the temporal shift requires models' capability for out-of-distribution (OoD) generalization [30], i.e., extrapolating from training environments to new unseen environments in remote future. Prior art that focus on model learning and evaluation over in-distribution data may yield sub-optimal results on OoD testing instances. *Second*, as mentioned before, there exist external factors that impact the generation of events. These external factors, which we term as *contexts*, are unobserved in practice. To eliminate their effects, one may also need the latent distributions that characterize how the contexts affect events generation. Unfortunately, such information is often inaccessible due to constrained data collection, which requires the models to learn from pure observed sequences.

## 1.1 Our Contributions

To resolve these difficulties, in this paper, we adopt a generative perspective to investigate the *temporal distribution shift* problem and propose a new variational context adjustment approach with instantiations to solve the issue for sequential event prediction.

**A generative perspective on temporal distribution shift.** We use proof-of-concept Structural Causal Models (SCMs) to characterize the dependency among contexts, historical sequences and labels (i.e., next event type) in terms of both data generation and model prediction. We show that contexts essentially act as a confounder, which leads the model to leverage spurious correlations and fail to generalize to data from new distributions. See Fig. 1 for a toy example to illustrate the issue.

**Variational context adjustment.** We propose a variational context adjustment approach to resolve the issue. The resulting new learning objective has two-fold effects: 1) helping to uncover the underlying relations between latent contexts and sequences in a data-driven way, and 2) canceling out the confounding effect of contexts to facilitate the model to explore the true causality of interest (as illustrated in Fig. 1). We also propose a mixture of variational posteriors to approximate context prior via randomly simulating pseudo input sequences.

**Instantiation by a flexible framework.** We propose a framework named as CaseQ to instantiate ingredients in the objective, which could combine with most off-the-shelf sequence backbone models. To accommodate temporal patterns under different environments, we devise a novel hierarchical branching structure for learning context-specific representations of sequences. It could dynamically evolves its architecture to adapt for variable contexts.

**Empirical results.** We carry out comprehensive experiments on three sequential event prediction tasks with valuation protocols designed for testing model performance under temporal distribution shift. Specifically, when we enlarge time gap between training and testing data, CaseQ can alleviate performance drop by $47.77\%$ w.r.t. Normalized Discounted Cumulative Gain (NDCG) and $35.73\%$ w.r.t. Hit Ratio (HR) for sequential recommendation, which shows its robustness against temporal distribution shift.

## 1.2 Related Works

**Sequential Event Prediction** [25] aims to predict the next event given a historical event sequence with applications including item recommendation in online commercial platforms, user behavior prediction in social networks, and symptom prediction for clinical treatment. Early works [39; 19; 18] rely on Markov chains and Bayesian networks to learn event associations. Later, deep learning based methods are proposed to capture non-linear temporal dynamics using CNN [45], RNN [20; 27; 54], attention models [24; 50; 51], etc. These models are usually trained offline using event sequences collected during a certain period of time. Due to temporal distribution shifts, they may not generalize well to online environments [41; 36; 22]. Our work develops a principled approach for enhancing sequential event prediction with better robustness against temporal distribution shifts via pursing causality behind data.

**Out-of-Distribution Generalization** [30; 4; 31; 63] deals with distinct training and testing distributions. It is a common yet challenging problem, and has received increasing attention due to its significance [26; 42; 52; 55]. For event prediction, distribution shift inherently exists because of different time intervals from which training and testing data are generated. Most existing sequence learning models assume that data are independent and identically distributed [20; 14; 37]. Despite various approaches of OoD generalization designed for machine learning problems in other research areas (e.g., vision and texts), it still remains under-explored for sequential event prediction and sequential recommender systems.

**Causal Inference** [33; 35] is a fundamental way to identity causal relations among variables, and pursuit stable and robust learning and inference. It has received wide attention and been applied in various domains e.g., computer vision [57; 28], natural language processing [40; 32] and recommender systems [58; 1]. Some existing causal frameworks for user behavior modeling [38; 16; 60; 61; 17] aim to extract causal relations based on observed or predefined patterns, yet they often require domain knowledge or side information for guidance and also do not consider temporal distribution shift. [1; 2; 6; 48] adopt counterfactual learning to overcome the effect of an ad-doc bias in recommendation task (e.g., exposure bias, popularity bias) or mitigate clickbait issue, whereas they do not focus on modeling sequential events and are still based on MLE as learning objectives.

## 2 Problem and Model Formulation

We denote $\mathcal{X} = \{1, 2, \cdots, M\}$ as the space of event types, and assume each event is assigned with an event type $x_i \in \mathcal{X}$. Events occurred in chronological order consist of a sequence $\mathcal{S} = \{x_1, x_2, \cdots, x_{|\mathcal{S}|}\}$. As mentioned before, the data distribution is normally affected by time-dependent external factors, i.e., *context c*. We use $S, Y, \hat{Y}$ and $C$ to denote the random variables of historical event sequence $\mathcal{S}$, ground-truth next event $y$, predicted next event $\hat{y}$ and context $c$, respectively. The data distribution can be characterized as $P(S, Y|C) = P(S|C)P(Y|S, C)$.

**Problem Formulation.** Given training data $\{(\mathcal{S}_i, y_i)\}_{i=1}^N$ generated from data distributions with $(\mathcal{S}_i, y_i) \sim P(S, Y|C = c_{tr}^{(i)})$, where $c_{tr}^{(i)}$ denotes the specific context when the $i$-th training sample is generated, we are to learn a prediction model $\hat{y}_i = f(\mathcal{S}_i; \theta)$ that can generalize to testing data $\{(\mathcal{S}_j, y_j)\}_{j=1}^{N'}$ from new distributions with $(\mathcal{S}_j, y_j) \sim P(S, Y|C = c_{te}^{(j)})$, where $c_{te}^{(j)}$ denotes the specific context when the $j$-th testing sample is generated. The distribution shift stems from different contexts that change over time, which we call *temporal distribution shift* in this paper.

### 2.1 Understanding the Limitations of Maximum Likelihood Estimation

Most of existing approaches target maximizing the likelihood $P_\theta(y|\mathcal{S})$ as the objective for optimization. Here we use $P_\theta(\cdot)$ to denote the distribution induced by prediction model $f_\theta$. Based on the definitions, we can build two Structural Causal Models (SCMs) that interpret the causal relations among 1) $C, S, Y$ (given by data-generating process) and 2) $C, S$ and $\hat{Y}$ (given by model learning), as shown in Fig. 2(a) and (b).

For Fig. 2(a), the three causal relations are given by the definitions for data generation, i.e., $P(S, Y|C) = P(S|C)P(Y|S, C)$. We next illustrate the rationales behind the other two causal relations $S \to \hat{Y}$ and $C \to \hat{Y}$.

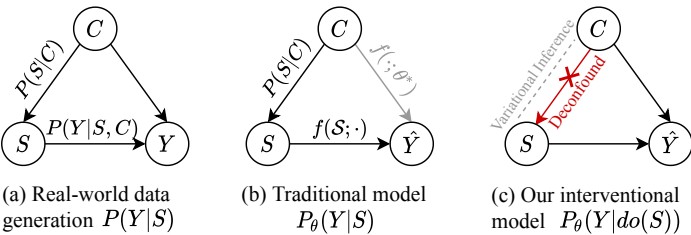

Figure 2: Structural causal model for sequence learning.

$S \rightarrow \hat{Y}$: This relation is induced by the prediction model $\hat{y} = f(\mathcal{S}; \theta)$ that takes a historical event sequence $\mathcal{S}$ as input and outputs the prediction for next event $\hat{y}$. The relation from $\mathcal{S}$ to $\hat{y}$ is deterministic given fixed model parameters $\theta$.

$C \rightarrow \hat{Y}$: This relation is implicitly embodied in the learning process that optimizes the model parameters with a given dataset collected at one time. By our definition, the training dataset is generated from a latent distribution affected by context $c_{tr}$ and the MLE algorithm yields

$$\theta^* = \arg\min_{\theta} \mathbb{E}_{(\mathcal{S},y) \sim P(S,Y|C=c_{tr})}[l\left(f(\mathcal{S};\theta), y\right)], \tag{1}$$

where $l(\cdot, \cdot)$ denotes a certain loss function, e.g., cross-entropy. This indicates that the learned model parameters $\theta^*$ is dependent on the distribution of $c_{tr}$. Also, due to the fact $\hat{y} = f(\mathcal{S}; \theta)$, we conclude the proof for relation from $C$ to $\hat{Y}$. In short, intuitive explanations for such a causal relation lie in two facts: 1) $C$ affects the generation of data used for model training, and 2) $\hat{Y}$ is the output of the trained model given input sequence $S$.

**Confounding Effect of $C$.** The key observation is that $C$ acts as the confounder in both Fig. 2(a) and (b), which could play a crucial role leading to undesirable testing performance of MLE-based approaches once the distribution is shifted. As implied by the causal relations in Fig. 2(a), there exists partial information in $S$ that is predictive for $Y$ yet highly sensitive to $C$, a usually fast-changing variable in real-world scenarios. As a result, the correlation between $S$ and $Y$ in previous contexts may become spurious in future ones. This also explains the failure of MLE-based models for generalizing to data from new distributions, according to the similar causal pattern in Fig. 2(b).

We can reuse the toy example in Fig. 1 as an intuitive interpretation for the failure. The 'summer' season (a context) acts as a confounder that correlates buying 'ice cream' (a historical event) and buying 'T-shirt' (a label), between which exists no obvious causal relation. However, the model would 'memorize' their correlation and tend to improperly recommend 'T-shirt' in the 'winter' season (new context). Whereas in fact the user purchases ice cream in winter most possibly because he/she is a dessert lover, in which case recommending other desserts would be a better decision.

**Intervention.** To address the confounding effect of $C$ and endow the model with robustness to temporal distribution shift, we propose to target model learning with $P_{\theta}(Y|do(S))$ instead of the conventional $P_{\theta}(Y|S)$. As shown in Fig. 5(c), the $do$-operator cuts off the arrow (i.e., causal relation) coming from $C$ to $S$, which essentially simulates an ideal data-generating process where sequences are generated independently from contexts. This operation blocks the backdoor path $S \leftarrow C \rightarrow \hat{Y}$ that spuriously correlates $S$ and $Y$, and enables the model to learn the desired causal relation $S \rightarrow Y$ which is invariant to environmental change.

## 2.2 Variational Context Adjustment

An ideal way to compute $P_{\theta}(Y|do(S))$ is to carry out randomized controlled trial (RCT) [34] by recollecting data from a prohibitively large quantity of random samples under any possible context, which is infeasible since we could neither control the environment nor collect data in the future. Fortunately, there exists a statistical estimation of $P_{\theta}(Y|do(S))$ by leveraging backdoor adjustment [34], wherein the confounder $C$ is stratified into discrete pieces $\mathcal{C} = \{c_i\}_{i=1}^{|\mathcal{C}|}$. By using

basic rules induced by the $do$-operator (see derivation in Appendix A), we have:

$$P_\theta(Y|do(S)) = \sum_{i=1}^{|\mathcal{C}|} P_\theta(Y|S, C = c_i)P(C = c_i). \tag{2}$$

Intuitively, the backdoor adjustment approximates an ideal situation where $c_i$ is enumerated according to the context prior $P(C)$ independent from input sequences, which serves to counteract the effect of $C$ on the generation of $S$. Nevertheless, optimizing with Eq. (2) is intractable since $c_i$ is usually unobserved or even undefined, and its prior distribution $P(C)$ is also unknown. Furthermore, computing Eq. (2) requires awareness of the relation between contexts and generated sequences, which is implicit in the data-generating process behind the data.

To address the difficulty for targeting Eq. (2), we introduce a variational distribution $Q(C|S)$ as the estimation for latent contexts given input sequences. By treating $C$ as a latent variable and using variational inference technique, we can obtain the following tractable evidence lower bound (ELBO) as learning objective (see derivation in Appendix B):

$$\begin{aligned}
&\log P_\theta(Y|do(S = \mathcal{S})) \\
\geq & \mathbb{E}_{c \sim Q(C|S=\mathcal{S})} \left[\log P_\theta(Y|S = \mathcal{S}, C = c)\right] - \mathcal{D}_{KL}\left(Q(C|S = \mathcal{S})\|P(C)\right),
\end{aligned} \tag{3}$$

where the last step is given by Jensen's Inequality and the equality holds if and only if $Q(C|S)$ exactly fits the true posterior $P(C|S, Y)$, which suggests it successfully uncovers the latent context from observed data. The first component in Eq. (3) is the negative reconstruction error (i.e. prediction error). The second component is the Kullback–Leibler (KL) divergence of the variational distribution and context prior distribution. Similar with [29], we could re-write the second term as a summation of the entropy of $Q(C|S)$, i.e., $\mathbb{H}[Q(C|S = \mathcal{S})]$ and the cross-entropy between $Q(C|S)$ and prior distribution, i.e., $\mathbb{H}[Q(C), P(C)]$. The former enforces high variance of contexts for each event sequence, and the latter aligns the aggregated posterior with the prior.

**Learnable Prior via Mixture of Posteriors.** The distribution $P(C)$ characterizes the true context prior in real world. It serves to regularize $Q(C|S)$ through the KL term in Eq. (3). Choosing an appropriate $P(C)$ is important yet challenging. One straightforward solution adopted by some previous works [8; 57] is to use a pre-defined prior such as uniform distribution. However, such a simplistic distribution may not reflect the true context prior, and potentially lead to over-regularization [5]. An alternative way is to estimate the prior by computing the average of the variational posterior [21], i.e., $P(C) \approx 1/N \sum_{i=1}^{N} Q(C|S = \mathcal{S}_i)$. However, such a method is computationally expensive [46] and would often result in biased estimation given a limited quantity of training data collected within a certain time interval. Inspired by [46] using mixture of Gaussian as a flexible and learnable prior, we propose to use a mixture of pseudo variational posteriors as an estimation:

$$\hat{P}(C) = \frac{1}{R} \sum_{j=1}^{R} Q(C|S = \mathcal{S}_j'). \tag{4}$$

where $\mathcal{S}_j'$ is a randomly generated pseudo event sequence and we set $R \ll N$ to reduce the computational cost. Note that the variational distribution $Q(C|S)$ is given by the prediction model (see details in Section 3). Therefore, the prior estimation by Eq. (4) is a general reflection of how the model favors each context given uninformative inputs. Also, the estimated prior is learned with the model in a fully data-driven manner.

## 3 Model Instantiations

We next parameterize the components in Eq. (3), i.e., $P_\theta(Y|S, C)$ and $Q(C|S)$, and propose a flexible framework named as CaseQ by taking 'cas' from 'causal' and 'seq' from 'sequence'.

**Event Embedding.** Given an event sequence $\mathcal{S} = \{x_1, x_2, \cdots, x_t\}$ as input, we use an embedding to represent each type of event. We consider a global embedding matrix $\mathbf{H}_x \in \mathbb{R}^{M \times d}$ to map each type of event into a $d$-dimensional embedding space:

$$\mathbf{h}_m^1 = OneHot(x_m)^\top \mathbf{H}_x, \tag{5}$$

where $M$ denotes the number of event types, $OneHot(\cdot) : \mathbb{Z}^+ \to \{0, 1\}^M$ transforms an event into a one-hot column vector, and $\mathbf{h}_m^1 \in \mathbb{R}^d$ denotes the initial representation for an event $x_m$.

**Inference Unit: Context-Specific Encoder.** To encode input sequences and accommodate the effect of $C$ on prediction, we consider an encoder $\Phi(\cdot)$ conditioned on a specific context. It takes a sequence of event embeddings and estimated context representation $\mathbf{c}_{(t)}$ at time step $t$ (given by the branching unit detailed next) as input, and outputs a sequence of hidden states:

$$[\mathbf{h}_1^2, \mathbf{h}_2^2, \cdots, \mathbf{h}_t^2] = \Phi([\mathbf{h}_1^1, \mathbf{h}_2^1, \cdots, \mathbf{h}_t^1], \mathbf{c}_{(t)}; \Theta), \tag{6}$$

where $\mathbf{h}_m^2$ is the $m$-th hidden state for the event sequence. Assume that there are $K$ types of contexts, i.e., $\mathcal{C} = \{\mathbf{c}_k\}_{k=1}^K$ and $c_{(t)} \in \mathcal{C}$. Each type of context $c_k$ is represented as a $K$-dimensional one-hot column vector $\mathbf{c}_k \in \{0,1\}^K$ whose $k$-th dimension is 1. Based on the SCM in Fig. 2, the context would change the underlying mechanism of how the next event is generated in real-world. Therefore, we consider context-specific *inference units* $\{\Phi_k(\cdot\,; \theta_k)\}_{k=1}^K$ as sub-networks of the encoder $\Phi(\cdot)$ to learn context-aware sequence representations, where $\Theta = \{\theta_k\}_{k=1}^K$. Then, we have

$$\Phi([\mathbf{h}_1^1, \mathbf{h}_2^1, \cdots, \mathbf{h}_t^1], \mathbf{c}_{(t)}; \Theta) = \sum_{k=1}^K \mathbf{c}_{(t)}[k] \cdot \Phi_k([\mathbf{h}_1^1, \cdots, \mathbf{h}_t^1]; \theta_k), \tag{7}$$

where $\mathbf{c}_{(t)}[k]$ returns the $k$-th entry of the vector $\mathbf{c}_{(t)}$. The inference unit can be specified as arbitrary off-the-shelf sequence models such as recurrent neural network (RNN) or self-attention (SA) [47].

**Branching Unit: Dynamic Model Selection.** We next introduce a *branching unit* $\Psi(\cdot)$ to parameterize the variational posterior $Q(C|S)$ that aims to obtain $\mathbf{c}_{(t)}$ given the sequence. At each time step, it takes the sequence representation as input and outputs a probability vector $\mathbf{q}_t \in [0,1]^K$, whose $k$-th entry represents the probability of the corresponding context $c_k$, i.e.,

$$\mathbf{q}_t = \Psi(\mathbf{h}_t^1; \Omega), \text{ where } \sum_{k=1}^K \mathbf{q}_t[k] = 1. \tag{8}$$

We use a $d'$-dimensional *context embedding* $\mathbf{w}_k \in \mathbb{R}^{d'}$ to accommodate the information of each context $\mathbf{c}_k$ by an embedding matrix $\mathbf{H}_c \in \mathbb{R}^{K \times d'}$, where $\mathbf{w}_k = \mathbf{c_k}^\top \mathbf{H}_c$ and $d' = d(d+1)$. Then, we split each context embedding into several fix-sized parameters $\mathbf{W}_k \in \mathbb{R}^{d \times d}, \mathbf{a}_k \in \mathbb{R}^d$:

$$\mathbf{W}_k = \mathbf{w}_k[:d^2].reshape(d,d), \ \mathbf{a}_k = \mathbf{w}_k[d^2 : d(d+1)]. \tag{9}$$

The attribution score $s_{tk}$ that measures how likely a sequence up to time step $t$ belongs to context $c_k$ can be calculated via

$$s_{tk} = \langle \mathbf{a}_k, Tanh(\mathbf{W}_k \mathbf{h}_t) \rangle. \tag{10}$$

Namely, we first project the sequence representation into a new $d$-dimensional space and then use dot product to measure the similarity. The variational distribution $Q(C|S = \mathcal{S})$ can be specified as a probability vector $\mathbf{q}_t$ that is computed by using Softmax function over $s_{tk}$, i.e., $\mathbf{q}_t = Softmax([s_{tk}]_{k=1}^K; \tau)$, where $\tau$ controls the confidence level. To implement the random sampling procedure $c \sim Q(C|S)$ in Eq. (3), one can also apply the categorical reparameterization trick that uses differentiable samples from Gumbel-Softmax [23] distribution, i.e.,

$$\mathbf{q}_t[k] = \frac{\exp\left((s_{tk} + g)/\tau\right)}{\sum_{k=1}^K \exp\left((s_{tk} + g)/\tau\right)}, \quad g \sim Gumbel(0,1). \tag{11}$$

Then, we can revise the right term of Eq. (7) as $\sum_{k=1}^K \mathbf{q}_t[k]\Phi_k([\mathbf{h}_1^1, \mathbf{h}_2^1, \cdots, \mathbf{h}_t^1]; \theta_k)$.

**CaseQ: a Hierarchical Branching Structure.** The aforementioned single-layered architecture requires an independent inference unit for each type of context, i.e., $|\mathcal{C}| = K$, which has several drawbacks: 1) It is impractical to accommodate a large amount of context types due to limited computational resources; 2) Real-world contexts are not entirely isolated, and thus independently parameterizing them may lead to over-fitting and undesired generalization performance given limited training data. To address these limitations, we further devise a hierarchical branching structure for CaseQ framework. Instead of using a one-hot vector, we denote a context as a 0-1 matrix, i.e.,

$$\mathbf{c}_k = stack([\mathbf{c}_k^1, \mathbf{c}_k^2, \cdots, \mathbf{c}_k^D]) \in \{0,1\}^{D \times K}, \tag{12}$$

where $D$ denotes the number of layers and $K$ denotes the number of parallel inference units in each layer. Each row in $\mathbf{c}_k$ is a one-hot vector indicating a certain inference unit in a layer, and each type of context corresponds to a certain combination of inference units. The entire network could then be represented as the stack of multiple parallel inference units and branching units:

$$[\mathbf{h}_1^{l+1}, \mathbf{h}_2^{l+1}, \cdots, \mathbf{h}_t^{l+1}] = \sum_{k=1}^{K} \mathbf{q}_t^l[k] \; \Phi_k^l([\mathbf{h}_1^l, \mathbf{h}_2^l, \cdots, \mathbf{h}_t^l]; \theta_k), \tag{13}$$

where $\mathbf{q}_t^l = \Psi^l(\mathbf{h}_t^l)$, and $l \in \{1, 2, \cdots, D\}$. The final prediction result at each time step can be obtained by ranking the relevance scores of current hidden states and event embeddings:

$$\hat{y}_t = \text{argmax}_{m \in \{1, \cdots, M\}} (\mathbf{H}_x \cdot \mathbf{h}_t^{D+1})[m]. \tag{14}$$

The above computation acts as a realization of $P_\theta(Y|S = \mathcal{S}, C = c)$ and $c \sim Q(C|S = \mathcal{S})$, required by the first term in Eq. (3). Besides, the KL term in Eq. (3) also requires density estimation from $Q(C|S = \mathcal{S})$. To achieve this, we use the continued Kronecker product of the probability vector $\mathbf{q}_t^l$ in each layer to represent the final variational posterior:

$$\mathbf{q}_t = Flatten\Big(\bigotimes_{l=1}^{D} \mathbf{q}_t^l\Big) \in \mathbb{R}^{K^D}, \tag{15}$$

where $\mathbf{q}_t^l \in \mathbb{R}^K$ and $\otimes$ denotes Kronecker product. The produced variational posterior is a $K^D$-dimensional vector, where each entry is the probability of a certain path (i.e., a combination of inference units across layers) in the network. This design has some merits. First, we could accommodate $|\mathcal{C}| = K^D$ types of contexts that grow exponentially w.r.t. model depth $D$. This helps to endow the model with larger capacity for learning diversified contexts. Second, different contexts may associate with each other by sharing the same inference unit in some layers, which plays as an effective *inductive bias* that guides the model to generalize to new environments (see more details and justifications in Appendix C.

According to Eq. (3), the final loss function can be written as

$$\sum_{\mathcal{S}} \sum_{t=1}^{|\mathcal{S}|-1} \left[ l(y_t, \hat{y}_t) + \alpha \mathcal{D}_{KL}\left( \mathbf{q}_t \| \frac{1}{R} \sum_{j=1}^{R} \mathbf{q}(\mathcal{S}'_j) \right) \right]. \tag{16}$$

where $\alpha$ is a weight parameter for balance, $\mathbf{q}(\mathcal{S}'_j)$ denotes the produced variational posterior distribution with a pseudo sequence $\mathcal{S}'_j$ as input, and $l(\cdot, \cdot)$ can be arbitrary loss functions (e.g., cross-entropy) depending on specific tasks.

## 4 Experiments

The goal of our experiments is to test whether CaseQ can effectively handle temporal event prediction under distribution shift and we consider several temporal prediction tasks with particular evaluation protocols for the purpose. The codes are available at `https://github.com/chr26195/Caseq`.

**Datasets.** We evaluate the effectiveness of CaseQ with different applications for event prediction, including sequential recommendation, ATM maintenance and user awarded badges prediction. We use four datasets with variable length and number of event types: *Movielens*, *Yelp*, *Stack Overflow* and *ATM*. The detailed description of datasets and their statistics are deferred to Appendix E.

**Evaluation Protocol.** The commonly adopted *leave-one-out* evaluation for sequential event prediction [24; 44; 45] uses the latest event for testing, the event right before the last for validation, and the rest for training. To test the effectiveness of CaseQ for handling temporal distribution shift, we generalize such evaluation protocol by enlarging the time gap between training and testing data. We use the last $G + 1$ events for testing, the first $|\mathcal{S}| - G - 2$ events for training, and the $(|\mathcal{S}| - G - 1)$-th event for validation. We call $G$ *maximal gap size* and $g \in [0, G]$ *gap size* which means the number of events between the validation event and a specific testing event. When $G = 0$ and $g = 0$, it degrades to the conventional leave-one-out evaluation. By increasing $g$ in our case, we could enlarge the time gap between training and testing data, so as to simulate temporal shift in real-world scenarios where data collection and online serving are during different periods of time. More details about the evaluation protocol are in Appendix G.

| | Gap Size | GRU4Rec [20] | | CaseQ (GRU) | | SASRec [24] | | CaseQ (SA) | | SSE-PT [49] | | CaseQ (PT) | |
|---|---|---|---|---|---|---|---|---|---|---|---|---|---|
| | | NDCG | HR | NDCG | HR | NDCG | HR | NDCG | HR | NDCG | HR | NDCG | HR |
| Movielens | 0 | 0.436 | 0.730 | 0.448 (+1.1%)[1] | 0.736 (+0.8%) | 0.453 | 0.742 | 0.462 (+2.0%) | 0.752 (+1.3%) | 0.465 | 0.753 | 0.476 (+2.3%) | 0.763 (+1.3%) |
| | 30 | 0.371 | 0.603 | 0.406 (+8.7%) | 0.660 (+8.7%) | 0.384 | 0.627 | 0.429 (+10.5%) | 0.675 (+7.1%) | 0.397 | 0.642 | 0.443 (+10.3%) | 0.680 (+5.5%) |
| | Drop (%) | ↓15.0 | ↓17.4 | ↓9.3 (−38.5%)[2] | ↓10.3 (−40.7%) | ↓15.2 | ↓15.6 | ↓7.1 (−53.1%) | ↓9.3 (−40.6%) | ↓14.6 | ↓14.7 | ↓7.0 (−51.8%) | ↓10.9 (−25.9%) |
| Yelp | 0 | 0.428 | 0.714 | 0.440 (+2.8%) | 0.731 (+2.3%) | 0.436 | 0.734 | 0.452 (+3.7%) | 0.744 (+1.4%) | 0.451 | 0.745 | 0.462 (+2.3%) | 0.751 (+0.9%) |
| | 20 | 0.395 | 0.663 | 0.426 (+7.2%) | 0.702 (+5.5%) | 0.402 | 0.682 | 0.433 (+7.0%) | 0.719 (+5.2%) | 0.405 | 0.685 | 0.440 (+8.0%) | 0.717 (+4.5%) |
| | Drop (%) | ↓7.6 | ↓7.2 | ↓3.2 (−57.9%) | ↓4.1 (−43.1%) | ↓7.6 | ↓7.1 | ↓4.3 (−43.3%) | ↓3.4 (−52.5%) | ↓10.3 | ↓8.0 | ↓4.7 (−54.5%) | ↓4.5 (−43.7%) |

[1] The percentage measures the improvement of CaseQ over the counterpart baseline with respect to one metric (e.g., NDCG).
[2] The percentage measures the degree of CaseQ helping to alleviate the performance drop compared to the counterpart.

Table 1: Experiment results of CaseQ with different base models. The *Drop* measures the percentage of decrease of a metric when time gap size increases.

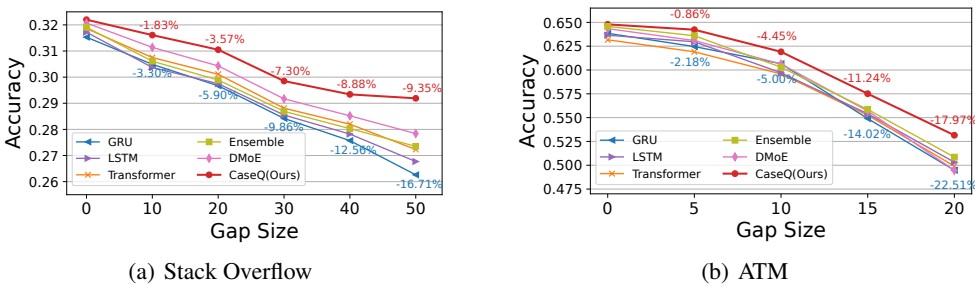

(a) Stack Overflow  (b) ATM

Figure 3: Performance comparison of CaseQ and baselines w.r.t. different gap sizes.

**Evaluation Metrics.** For recommendation datasets, i.e., Movielens and Yelp, we adopt two widely used evaluation metrics: *Hit Ratio at K* (HR@$K$), and *Normalized Discounted Cumulative Gain at K* (NDCG@$K$), where $K = 10$ in practice. For other two low-dimensional datasets with fewer number of event types, i.e., Stack Overflow and ATM, we use Accuracy (i.e., Precision) as the metric.

### 4.1 Sequential Recommendation

We adopt three sequence models as backbones for sequential recommendation: 1) GRU4Rec [20] uses RNNs to model users' sequential behaviors and predict the next clicked item; 2) SASRec [24] adopts Self-Attention (SA) in Transformer's encoder for sequential recommendation; 3) SSE-PT [49] further considers user's personalized embedding which incorporates user information into the input sequence. We use MLE for training these models. For fairness, our model uses the same number of layers as the baselines. We set gap size $g$ as $0, 30$ for Movielens and $0, 20$ for Yelp since the average sequence length of Movielens is greater than Yelp's. As shown in Table 1, our approach consistently outperforms the counterparts with different backbones throughout all the cases. While the performance degrades when we increase the gap size, CaseQ is significantly more robust to temporal distribution shift than conventional models. The percentage of the performance drop for CaseQ is is significantly lower than the counterpart model. Also, CaseQ is agnostic to the sequential prediction backbones and can be further combined with other advanced architectures.

### 4.2 Event Prediction

We further study the performance of CaseQ in Stack Overflow and ATM datasets, which have distinct statistical features compared to sequential recommendation datasets. We consider three commonly adopted models for predicting discrete sequential events, i.e., GRU, LSTM and Transformer [47]. We also consider two ensemble frameworks that has the same model size with ours and are built on parallel structures: 1) *Ensemble* [7] adopts mean pooling over the hidden states of different inference units; 2) *DMoE* [10] combines the hidden states by learning a gating network in each layer. Also,

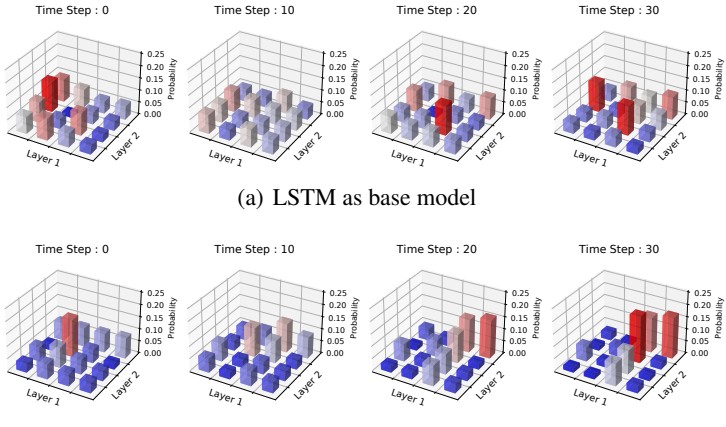

(a) LSTM as base model

(b) Transformer encoder as base model

Figure 4: Visualization of probabilities for different contexts recognized by well-trained CaseQ ($D = 2, K = 4, \tau = 1$). The results reveal the temporal distribution shift behind data.

we use multiple heads for Transformer to enlarge its model size. For Stack Overflow, the gap size varies from 0 to 50. For ATM, it varies from 0 to 20. As shown in Fig. 3, CaseQ achieves the best results throughout all the cases in both datasets. As the gap size increases, the overall accuracy of baseline models exhibit a sharp downward trend in general, which shows the severe negative impact of temporal distribution shift. Despite with larger model size, both ensemble-based baselines show little improvement over the single models and also suffer a serious performance drop with larger gap sizes. In comparison, when the gap size goes large, the performance improvement of CaseQ over other competitors becomes more significant. This suggests that our approach is indeed effective for improving the robustness of sequence models w.r.t. temporal shift.

### 4.3 Case Study

**Visualization of Temporal Shift.** To further investigate temporal distribution shift, in Fig. 4 we visualize the probabilities for each type of context at different time steps in a sequence. We set $K = 4, D = 2$. The two sub-figures show the results with different base models (LSTM and Transformer). The probability value reflects the intensity of a certain context at one time. As shown in the figure, while some contexts dominate at the beginning, they exhibit clear variation over time. The results imply that there indeed exists temporal distribution shift behind the data collected at different time intervals. Therefore, prior art use the past sequential data for training may not successfully transfer to new environments in the future. By contrast, our approach could first identifies latent contexts behind data and then alleviate its impact on learning a stable prediction model.

**Learning of Context Embeddings.** Context embeddings are latent representations for each type of context and dynamically affect the branching unit for selecting the proper inference unit. Conceptually, each type of context is expected to represent an environment in physical world, so each context embedding is supposed to be informative and distinctive. To verify whether our model could learn meaningful context representations without collapse or over-regularization, we visualize the context embeddings ($K = 3, D = 2$) through the training process in Fig. 5 by using dimensional reduction. Each color corresponds to the embedding of a certain context. While the context embeddings have the same initialized positions, they gradually separate and move towards different directions as the epochs increase. This indicates that the model manages to learn informative context representations.

### 4.4 Scalability Test

We further test the scalability of CaseQ w.r.t. the number of context types. Fig. 6 shows the training time and inference time per mini-batch of data. We use a single layer when the number of context types is within $[1, 6]$ and two layers when it further increases. As we can see in the range of $[1, 6]$, the training and inference time exhibit linear increase with a single-layer architecture. The increasing trend goes down when we increase the model depth. The results suggest CaseQ's decent scalability.

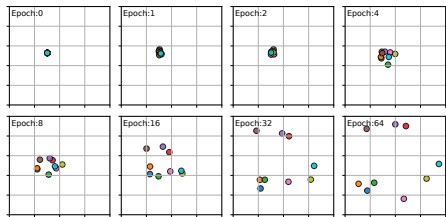

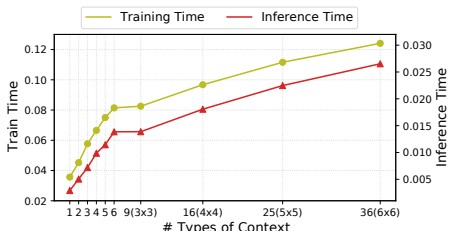

Figure 5: Visualization of context embeddings (reduced into a 2D space, $D = 3, K = 3$) at different training epochs.

Figure 6: Train and inference time (s) of CaseQ w.r.t. number of context types.

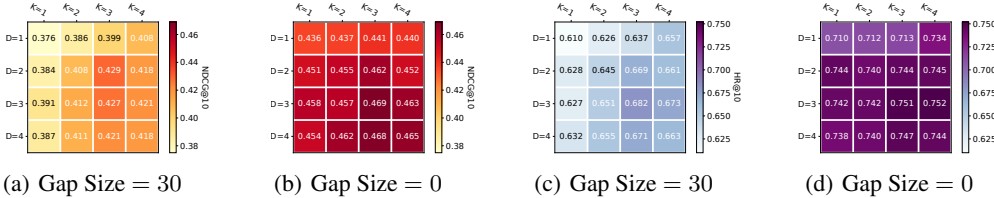

(a) Gap Size = 30          (b) Gap Size = 0          (c) Gap Size = 30          (d) Gap Size = 0

Figure 7: Performance of CaseQ on recommendation task w.r.t. different $K$ and $D$.

## 4.5 Hyper-Parameter Analysis

An appropriate number of context types are important for guaranteeing the performance and generalizability of our model. Recall that there are in total $K^D$ types of contexts, given a model whose depth (i.e. number of layers) is $D$ and width (i.e. number of parallel inference units in a layer) is $K$. Fig. 7 shows the performance of CaseQ w.r.t. different $K$ and $D$ in sequential recommendation task. The results show that increasing $K$ and $D$ in a certain range is effective for improving the performance. Notably, the improvements are more significant with larger gap size, which again verifies that CaseQ is helpful for addressing temporal shift. Still, out-of-range $K$ and $D$ would harm the performance, which is presumably due to over-fitting.

## 5 Limitations and Conclusion

**Limitations and Potential Impacts.** One of the main limitations of our variational context adjustment framework is that it stratifies discrete contexts (which may be infinitely many) into a fixed context set, in order to make the objective tractable. However, this remains a common issue for all works that are dealing with infinitely many discrete confounders, and our framework has already made a progress in this direction increasing the size of context set to $K^D$. We expect to see more works to be done in this direction. Also, due to the nature of the problem setting where the context is defined as a latent and abstract factor (which is a major challenge we aim to tackle and serves as our contribution), it would be hard or even impossible to identify its meaning in the real-world. Finally, we do not foresee any direct negative societal impacts.

**Conclusion.** In this paper, we endeavor to deal with temporal distribution shift in sequential event prediction. Namely, the training and testing data are generated from distinct distributions with different contexts. We first show that existing approaches with maximum likelihood estimation would leverage spurious correlation and produce undesirable prediction. Then we propose a new learning objective with backdoor adjustment and variation inference techniques. Based on this, we devise a novel model framework for learning context-specific representations. Extensive experiments demonstrate the practical efficacy of proposed method. We hope our work could enlighten new research focus for subsequent works and inspire more frontier research in sequential event prediction against distribution shift.

## Acknowledgement

This work was partly supported by National Key Research and Development Program of China (2020AAA0107600), National Natural Science Foundation of China (61972250, 72061127003), and Shanghai Municipal Science and Technology (Major) Project (2021SHZDZX0102, 22511105100).

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
