# A  Derivation of Backdoor Adjustment

We show the derivation of backdoor adjustment for the proposed causal graph in Fig. 2 based on the rules with $do$-calculus [33]. Consider a directed acyclic graph $\mathcal{G}$ with three nodes: $X$, $Y$ and $Z$. We denote $\mathcal{G}_{\overline{X}}$ as the intervened causal graph by cutting off all arrows coming into $X$, and $\mathcal{G}_{\underline{X}}$ as the graph by cutting off all arrows going out from $X$. For any interventional distribution compatible with $\mathcal{G}$, we have the following two rules:

**Rule 1.** *Action/observation exchange:*

$$P(y|do(x), do(z)) = P(y|do(x), z), \ if \ (Y \perp Z|X)_{\mathcal{G}_{\overline{X}\underline{Z}}} \tag{17}$$

**Rule 2.** *Insertion/deletion of actions:*

$$P(y|do(x), do(z)) = P(y|do(x)), \ if \ (Y \perp Z|X)_{\mathcal{G}_{\overline{X}\overline{Z}}} \tag{18}$$

We could derive the interventional distribution via:

$$
\begin{aligned}
P_\theta(Y|do(S = \mathcal{S})) &= \sum_{i=1}^{|\mathcal{C}|} P_\theta(Y|do(S = \mathcal{S}), C = c_i) P(C = c_i|do(S = \mathcal{S})) \\
&= \sum_{i=1}^{|\mathcal{C}|} P_\theta(Y|do(S = \mathcal{S}), C = c_i) P(C = c_i) \\
&= \sum_{i=1}^{|\mathcal{C}|} P_\theta(Y|S = \mathcal{S}, C = c_i) P(C = c_i),
\end{aligned}
\tag{19}
$$

where the first step follows the law of total probability, the second step applies Rule 2, and the last step applies Rule 1.

# B  Derivation of Variational Context Adjustment

We derive the variational context adjustment in Eq. (3). We re-write the logarithm of interventional distribution $P_\theta(Y|do(S = \mathcal{S}))$ as

$$\log \mathbb{E}_{c \sim P(C)} \left[ P_\theta(Y|S = \mathcal{S}, C = c) \right]. \tag{20}$$

We introduce a variational distribution $Q(C|S = \mathcal{S})$ as an approximation of the true posterior $P(C|S = \mathcal{S}, Y = y)$. By using variational inference, we derive a tractable lower bound as objective:

$$
\begin{aligned}
&\log P_\theta(Y|do(S = \mathcal{S})) \\
&= \log \mathbb{E}_{c \sim P(C)} \left[ P_\theta(Y|S = \mathcal{S}, C = c) \frac{Q(C = c|S = \mathcal{S})}{Q(C = c|S = \mathcal{S})} \right] \\
&= \log \mathbb{E}_{c \sim Q(C|S=\mathcal{S})} \left[ P_\theta(Y|S = \mathcal{S}, C = c) \frac{P(C = c)}{Q(C = c|S = \mathcal{S})} \right] \\
&\geq \mathbb{E}_{c \sim Q(C|S=\mathcal{S})} \left[ \log P_\theta(Y|S = \mathcal{S}, C = c) \frac{P(C = c)}{Q(C = c|S = \mathcal{S})} \right] \\
&= \mathbb{E}_{c \sim Q(C|S=\mathcal{S})} \left[ \log P_\theta(Y|S = \mathcal{S}, C = c) \right] - \mathcal{D}_{KL} \left( Q(C|S = \mathcal{S}) \| P(C) \right),
\end{aligned}
\tag{21}
$$

where the third step is given by Jensen Inequality.

# C  More Discussions

We gather the continued Kronecker product of the probability vector $\mathbf{q}_t^l$ in each layer to compute the final variational posterior $\mathbf{q}_t$:

$$\mathbf{q}_t = Flatten \left( \bigotimes_{l=1}^{D} \mathbf{q}_t^l \right) \in \mathbb{R}^{K^D}. \tag{22}$$

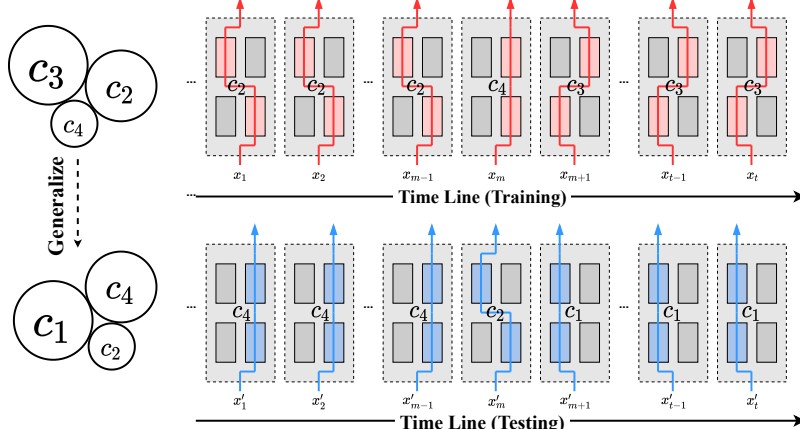

Figure 8: An intuitive explanation of how CaseQ generalize to novel context. Each small solid box denotes an inference unit. $c_k$ denotes the context type at a certain time step. The size of the bubble represents its frequency. The colored boxes (in red for training and blue for testing) highlight the inference units selected by branching models.

Kronecker product is a generalization of the outer product from vectors to matrices. The entries of $\mathbf{q}_t$ denotes the probability for every possible combination of inference units (termed as *path*) at time step $t$. For example, consider $D = 3, K = 3$. We have $\mathbf{q}_t^l \in \mathbb{R}^3$, $\mathbf{q}_t^1 \otimes \mathbf{q}_t^2 \otimes \mathbf{q}_t^3 \in \mathbb{R}^3 \times \mathbb{R}^3 \times \mathbb{R}^3$, and $\mathbf{q}_t \in \mathbb{R}^{27}$. The $i$-th element in $\mathbf{q}_t$ denotes the probability of a certain path in the network, representing the probability of a type of context $Q(C = c_i | S = \mathcal{S})$. Suppose $i = (xyz)_3$ is a ternary number, the path corresponds to the combination of $z/y/x$-th inference unit at the first/second/third layer.

**How CaseQ generalizes to novel context?** As mentioned before, one advantage of such a design is that it could associate different types of contexts through shared inference units in some layers. By this way, we build links between majority (seen) contexts and minority (unseen) contexts. It can guide the model to generalize to future environments when the minority context may become the major one or unseen context occurs. We provide an example in Fig. 8 for further illustration. In training stage, the latent contexts gradually change over time (and so as the corresponding path in the network). During this time, $c_2$ and $c_3$ are majority contexts, $c_4$ is the minority context, and $c_1$ is a unseen one. Despite the imbalanced contexts, each inference unit as a module is relatively equally trained. Therefore, in testing stage, when $c_1$ and $c_4$ become majority contexts, our model can still generalize well by using well-trained inference units as modules.

## D   Comparison with Existing Causal Models

We compare CaseQ with existing causal models for sequence learning and user behavior modeling. Some existing methods [38; 16; 60; 61; 17] aim to extract causal relations based on observed or predefined patterns, yet they often require domain knowledge or side information for guidance and also do not consider temporal distribution shift. [1; 2; 6; 48] adopt counterfactual learning to overcome the effect of an ad-doc bias in recommendation task (e.g., exposure bias, popularity bias) or mitigate clickbait issue. Differently, they do not focus on modeling sequential events and are still based on maximum likelihood estimation as learning objectives. There are also few recent works that attempt to adopt causal inference to de-noise input sequences in sequential user behaviors[58]. Yet, they overlook the distribution shift and have distinct research focus compared with this paper.

## E   Dataset Descriptions

**Movielens** Movielens [15] is a widely used benchmark dataset for sequential recommendation evaluation. It contains one million ratings of 3900 movies made by 6040 users. We use each user's rating records as implicit feedbacks and construct a sequence of rated movies for each user in the order of timestamps, where each rating for a movie is treated as an event.

| Datasets | #Sequences | #Event Types | #Events | Avg. length |
|---|---|---|---|---|
| Movielens | 6039 | 3415 | 1.00M | 165.50 |
| Yelp | 23056 | 15575 | 0.649M | 28.14 |
| Stack Overflow | 4777 | 22 | 0.345M | 72.25 |
| ATM | 1554 | 7 | 0.552M | 355.35 |

Table 2: Statistics of four datasets.

**Yelp**  Yelp is a dataset for business recommendation, provided by the Yelp Dataset Challenge [2]. After pre-processing and filtering, it contains $648, 687$ reviews for businesses by $23, 057$ users. We treat all user reviews as events and sort them chronically into sequences.

**Stack Overflow**  This dataset is collected from a question answering website Stack Overflow [9]. It includes $480, 000$ records of awarded badges of $6, 000$ users during two years between 2012 and 2014. Each awarded badge is treated as an event. The task is to predict the next awarded badge of users.

**ATM**  This dataset is collected from 1554 ATMs owned by an anonymous global bank headquartered in North America [53], which includes 7 types of events including 6 error types. It has a total number of $552, 215$ events. The task is to predict future events for better maintenance support services.

The statistics of these datasets are summarized in Table 2.

# F    Implementation Details

We implement CaseQ with PyTorch. All parameters are initialized with Xavier initialization method. We train the model by Adam optimizer. All the models are trained from scratch without any pre-training on GTX 2080 GPU with 11G memory. For comparative methods, we refer to the hyper-parameter settings in their papers and also finetune them on different datasets. For sequences whose length is greater than 100, we only consider the most recent 100 events to speed up training. We run all experiments 5 times and take the average value.

# G    Evaluation Protocol and Problem Setting

Given a dataset consists of $N$ event sequences, i.e., $\mathcal{D} = \{\mathcal{S}_i\}_{i=1}^{N}$, where $\mathcal{S}_i = \{x_{i,1}, x_{i,2}, \cdots, x_{i,|\mathcal{S}_i|}\}$. We use $\mathcal{S}_i[: t]$ to denote the first $t$ events in the sequence and $\mathcal{S}_i[-t :]$ to denote the last $t$ events. The training, validation and test sets are:

$$
\begin{aligned}
\mathcal{D}_{train} &= \{(\mathcal{S}_i[: t-1], x_{i,t})\}_{i\in\{1,\cdots,N\}, t\in\{2,\cdots,|\mathcal{S}_i|-G-2\}} \\
\mathcal{D}_{valid} &= \{(\mathcal{S}_i[: t-1], x_{i,t})\}_{i\in\{1,\cdots,N\}, t=|\mathcal{S}_i|-G-1} \\
\mathcal{D}_{test} &= \{(\mathcal{S}_i[: t-1], x_{i,t})\}_{i\in\{1,\cdots,N\}, t\in\{|\mathcal{S}_i|-G,\cdots,|\mathcal{S}_i|\}},
\end{aligned}
\tag{23}
$$

where $G$ is the maximal gap size. When the sequence length is less than or equal to $G + 3$, we use the whole sequence for training. Given a specific gap size $g \in [0, G]$, we evaluate the model performance on a test subset:

$$
\mathcal{D}_{test}(g) = \{(\mathcal{S}_i[: t-1], x_{i,t})\}_{i\in\{1,\cdots,N\}, t=|\mathcal{S}_i|-G+g} \quad . \tag{24}
$$

By increasing $g$, we could enlarge the time gap between training and testing data, so as to simulate temporal distribution shift in real-world scenario where data collection and online serving are during different periods of time, and evaluate model's effectiveness against distribution shift. We are interested in the model performance drop as a metric to evaluate the robustness to distribution shift. Note that evaluations with different $g$ are conducted simultaneously using the same model to control variates. When $G = 0$ and $g = 0$, this evaluation method exactly degrades to widely used leave-one-out evaluation protocol that is widely used in recommendation tasks [24; 44; 45].

---

[2]https://www.yelp.com/dataset/challenge