# OpenReview forum: "Towards Out-of-Distribution Sequential Event Prediction: A Causal Treatment"
_NeurIPS.cc/2022/Conference — NeurIPS 2022 Accept_

### Official Review · Reviewer_k7dp · 2022-07-10

**Rating:** 6
**Confidence:** 3
**Soundness:** 4 excellent
**Presentation:** 4 excellent
**Contribution:** 3 good

**Summary:**

The paper tackles the problem of event sequence modelling and forecasting by allievating problem of adapting to OOD data. The authors propose a variational learning algorithm that encodes the context embeddings which model changing data distribution of past event sequences and motivate it as a Structural Causal model where the causal relation between context and future even sequence is deconfounded so that the causal chain only goes through past event sequence.

**Questions:**

1. How does model deal with novel contexts?
2. Will the generative model adapt to uncertain situations or provide good uncertainty bounds?
3. EValuation on a benchmark where changes in context is known to test if the learned contexts map to domain knowledge would be useful.

**Limitations:**

1. .It is not clear how sensitive the model is to the number of contexts when dealing with datasets of differing scales and changes in distributions.
2. It is not clear how the model deals with novel or derivative contexts.
3. The underlying assumption of using backdoor adjustment to effectively deconfound context to events causal relationship may not in many real-world cases.

**Strengths And Weaknesses:**

Strengths
1. Paper is strongly motivated and tackles an important problem in deploying real-world predictive systems with rapidly changing data distributions.
2. The method is a novel application of SCM for event sequence prediction and modeling choices are well justified both rationally and empirically.
3. The empirical results show improvement in some standard benchmarks. Ablation and secondary results show that the model indeed automatically learns to detect changes in context and switches across them resulting in increase in performance.
Weaknesses
1. % improvement is used as the primary metric of comparison. A better approach would be to provide absolute results and note statistical significance.
2. Model doesn't adapt to novel context in test set and there is no evidence of good uncertainty estimation in unknown scenarios when facing novel contexts/data distributions.

---

> ### Author Response · Authors · 2022-08-02
> **Response to Reviewer k7dp (Part 1 of 2)**
>
> Thank you for your time and thorough review. We are glad that you appreciated our strong motivation, significance, novel method, and extensive experiments. In response to your questions and weakness points mentioned, we provide detailed answers to increase your confidence.
>
> > ***Q1: "How does model deal with novel contexts?"***
>
> This is indeed an intriguing question, and we consider two cases that differ in the definition of “novel context”. In a nutshell, our model could deal with novel contexts if they are in the context set supported by both intuitive and theoretical justifications provided in the following. We also claim the situation where novel context is not "in" the context set technically does not exist and discuss the real implication of "novel context" in our framework.
>
>
> **(Case 1)**. Novel context refers to a minority of unseen context in the context set, i.e., $c_{test} \in \mathcal C$.
>
> We have already provided some justifications for this case in Appendix C, and updated this part with more details and a new illustrative fig.8 for better understanding.
>
> In short, our hierarchical branching structure combined with the definition of context posterior is able to build connections between majority context and minority (or unseen) context. Therefore, our model is more likely being able to generalize to those novel contexts even if they are unseen in training. An intuitive but unrigorous example would be: Context 1 (seen in training) represents “summer and sunny”, context 2 (seen in training) represents “winter and rainy”, and we expect our model can generalize to context 3 (unseen in training) that represents “summer and rainy”.
>
>
> **(Case 2)**. Novel context refers to a context that is not in the context, i.e., $c_{test} \notin \mathcal C$.
>
> In practice, there probably exist infinitely many specific contexts and the objective is intractable in this case, which is why we **stratify context into discrete pieces** to create $\mathcal C$ (which is a conventional practice) as an approximation. In other words, each context now actually means "a class of specific contexts", and all contexts in $\mathcal C$ are sufficient to represent all existing specific contexts. In this sense, the size of $\mathcal C$ means **"to what extent of granularity we seperate context" or "the number of classes of context"** rather than "the number of specific context".
>
> Therefore, we agree in that **“using backdoor adjustment to effectively deconfound context to events causal relationship may not in many real-world cases”** due to the aforementioned approximation. This remains a common issue for all works that are dealing with infinitely many discrete confounders, and our framework has already made a progress in this direction by increasing the size to $K^D$.
>
> Handling novel (specific) context subsequently boils down to accurately predicting which class of contexts it belongs to. Since the prediction results of novel context presumably is less confident, it is indeed useful to consider uncertainty estimation in this case.
>
>
> > **Q2: “Will the generative model adapt to uncertain situations or provide good uncertainty bounds?”**
>
> We are not certain as we are unfamiliar with this field. But one possible solution would be using Q(C|S) produced by a well-trained model to compute a confidence score (e.g., directly based on softmax output or by measuring the difference with training data) for uncertainty estimation or out-of-distribution detection. The result might be useful for letting models to "know when they don't know" in the face of novel context (case 2), which could be an interesting future research direction.
>
>
>
> > ***Q3: “Evaluation on a benchmark where changes in context is known to test if the learned contexts map to domain knowledge would be useful.”***
>
> Considering the pressing time, we conducted an additional experiment based on a synthetic dataset generated by four different models trained on different parts of Yelp dataset for a quick (but not rigorous) verification. We manipulate the temporal distribution shift by treating two of these models as majority for generating training sequences and minority for testing sequences. We then re-train a model on the synthetic dataset with $K=4$. We compare the true and estimated frequency of context (where their correspondences are manually chosen) in the testing set, and the results are as follows.
>
> |  | Context 1 | Context 2 |Context 3 | Context 4 |
> | -------- | -------- | -------- |-------- | -------- |
> | Ground-Truth     | 0.4     | 0.4     | 0.1 | 0.1 |
> | Estimated     | 0.33     | 0.39     | 0.15 | 0.13|
>
> We will consider rigorous evaluation in other benchmarks in the future.

---

> > ### Author Response · Authors · 2022-08-02
> > **Response to Reviewer k7dp (Part 2 of 2)**
> >
> > >***Q4: "It is not clear how sensitive the model is to the number of contexts when dealing with datasets of differing scales and changes in distributions."***
> >
> > We have studied how the context number (determined by choices of $K$ and $D$) affect model's ability in dealing with temporal distribution shift in section 4.5 with fig.7. By comparing model's performance in different gap size, we observe that in general larger context number within certain range indicates lower performance drop.
> >
> > >***Q5: "% improvement is used as the primary metric of comparison. A better approach would be to provide absolute results and note statistical significance."***
> >
> > Thanks for the nice suggestion. We use % performance drop mainly because it is more intuitive and will try the method suggested by the reviewer in the future.

---

> ### Author Response · Authors · 2022-08-06
> **A kind reminder before the discussion phase ends**
>
> Dear reviewer k7dp,
>
> Thanks again for your review. Since the discussion period is approaching its end, we would be glad to hear from you if we have addressed your questions/concerns.
>
> Kind regards, The Authors

---

> > ### Comment · Reviewer_k7dp · 2022-08-09
> > **Thanks for the reply**
> >
> > I have read the reply to my questions as well as other reviews. I will keep the same score (6).

---

### Official Review · Reviewer_HB3W · 2022-07-12

**Rating:** 7
**Confidence:** 2
**Soundness:** 3 good
**Presentation:** 3 good
**Contribution:** 4 excellent

**Summary:**

The authors propose the CaseQ framework for temporal event prediction under distribution shift. They identify a clear problem in typical temporal event prediction approaches: learning spurious correlations between items in the training set that do not generalize to the test set, which consists of events at later times. They hypothesize that these spurious correlations are due to latent contexts that may change from the training to the test set. They consider structural causal models for the data generating process that incorporate context and propose to use a backdoor adjustment to deconfound the context. They then propose a variational approximation that is tractable and treat the evidence lower bound (ELBO) as the learning objective. They demonstrate that their CaseQ framework can be used to mitigate accuracy drops when the gap between the training and test sets grows over time.


**Questions:**

1. Which components of the model instantiations section are novel?
2. The proposed framework considers a very specific reason for distribution shift--dependence on a hidden context. What happens if there are distribution shifts for other reasons, e.g. changes in user preferences over time that may not be related to a context?


**Limitations:**

Limitations are not really discussed aside from some effects of hyperparameters in Section 4.5. There could be many different reasons for temporal distribution shift, and the proposed work considers one specific reason: dependence on a latent context. The discussion of limitations could be improved to discuss other potential reasons for context shift and how they could be incorporated into the model, or how their current model may try to capture these other reasons.


**Strengths And Weaknesses:**

Strengths:
- Addresses the distribution shift problem for temporal event prediction in an innovative manner (to the best of my knowledge) using a causal inference approach combined with variational inference.
- Strong experimental results showing that the proposed CaseQ framework can mitigate drops in recommendation accuracy by about 50% as the gap between training and test set grows to about 30 events.
- Paper is mostly well written and explains the reasoning behind the proposed generative perspective on temporal distribution shift in a friendly manner.

Weaknesses:
- Novelty of the specific model instantiations are unclear. It appears to me that the authors are taking standard sequence models (e.g. GRU) as the inference unit, but then then hierarchical branching structure is novel. It would be helpful to describe in the first paragraph of Section 3 which components are novel.
- Several of the figures are quite small (e.g. Figure 4) because the authors are trying to present a lot of results. I would suggest moving some of the results to the supplementary material, e.g. the hyperparameter analysis in Section 4.5. I'm not sure what I'm supposed to understand from Figure 4 aside from seeing some changes over time.
- Code is not provided with the submission.

Other minor issues:
- "Prior art" and "prior work" are already plural, not "prior arts" and "prior works"
- Time unit is missing in Figure 6. Train time of 0.12 in what unit of time?

---

> ### Author Response · Authors · 2022-08-02
> **Response to Reviewer HB3W**
>
> Thank you for the thorough review and nice suggestions. We are glad that you like our approach. We next answer your questions to increase your confidence.
>
> >***Q1: "It would be helpful to describe in the first paragraph of Section 3 which components are novel?”***
>
> Indeed, the inference unit could be specified as standard building blocks in sequence models such as GRU, LSTM and self-attention, which gives our model a lot of freedom to be combined with existing sequence models. Besides, we also use the standard event embedding layer in Eq.(5), output layer for producing prediction results in Eq.(14), and standard prediction loss in the first term of Eq.(16). All the rest components in section could be considered as novel parts that are specifically designed based on our variational context adjustment framework in Section 2. They are summarized as follows:
>
> 1. We use each inference unit (instantiation of P(Y|S,C), Eq.(6) and Eq.(7)) as a context-specific encoder to learn sequence representations conditioned on a specific context type, where the context is sampled in inference unit (instantiation of Q(C|S), Eq.(9) and Eq.(10)) by computing its distribution based on similarity scores.
> 2. We further use Gumbel-Softmax trick (Eq.(11)) to make the sampling process in Eq.(3) differentiable, extend the model to a hierarchical branching structure (Eq.(12) and Eq.(13)) to incorporate more context types (i.e., K^D) under limited sources, and propose a new way (Eq.(15)) to compute variational posterior that adapts to the hierarchical structure. The final implementation of Eq.(3) is given by the loss function in Eq.(16).
>
> >***Q2: "What happens if there are distribution shifts for other reasons, e.g. changes in user preferences over time that may not be related to a context?"***
>
> This an interesting question and we would like to add more details to answer it.
>
> **Definition and concept of context.**
>
> We mentioned in the introduction that the concept of context refers to external factors that may impact the generation of events. Here, “external” here means any random variables that is not sequence $S$ and event $Y$ themselves. In other words, the definition of “context” is not restricted to the conventional meaning of “context” such as season and fashion trends that we use only as examples. It could also be interpreted as abstract “user preference” or other properties relating to events and user themselves if they are indeed affecting the generation process of events (i.e., match the data generation process in fig.2).
>
> **How our framework address distribution shift caused by user preference or other factors.**
>
> Theoretically, our framework is general enough to address distribution shift caused by user preference shift or any other factors as long as 1) they are affecting or causing the data generation and  2) the event sequence is informative enough for our model to explore the shift. Therefore, **the question really is if we can truly discover the user preference shift or shifts of other latent confounding factors hidden in the data**, which the challenging part. This challenge is tackled in our work by integrating variational inference into the causal interventional model. If our model is not able to address distribution shift caused by these factors, it could be due to improper implementation/training, the intrinsic limitation of context stratification (see limitation in appendix), the bottleneck of dataset, or that they are not affecting the data generation.
>
> > ***Q3: "Several of the figures are quite small (e.g. Figure 4) because the authors are trying to present a lot of results. I would suggest moving some of the results to the supplementary material."***
>
> Thanks for the nice suggestion. We will modify them accordingly in the final version to further improve readability.
>
>
> >**Other minior issues (typos)**
>
> All fixed in the new version. Thanks for pointing them out.

---

> > ### Comment · Reviewer_HB3W · 2022-08-09
> > **Thank you**
> >
> > I thank the authors for the detailed replies to reviewer comments. The clarification on the role of context was helpful for me to better understand the contribution.

---

> ### Author Response · Authors · 2022-08-06
> **A kind reminder before the discussion phase ends**
>
> Dear reviewer HB3W,
>
> Thanks again for your review. Since the discussion period is approaching its end, we would be glad to hear from you if we have addressed your questions/concerns.
>
> Kind regards, The Authors

---

### Official Review · Reviewer_t8MQ · 2022-07-13

**Rating:** 6
**Confidence:** 3
**Soundness:** 3 good
**Presentation:** 3 good
**Contribution:** 2 fair

**Summary:**

The paper investigates the the temporal distribution shift problem or sequential event prediction and propose a new variational context adjustment approach. The main idea of the approve leveraged variational inference to estimate and adjust for the casual contexts, treating them as latent variables to be learned by the model. Lastly, a hierarchical branching structure was proposed to accommodate large number of events  .

**Questions:**

Could the authors comment on the model's performance where the context shifted to a point where previously common events now become are and previously rare events now become common. The latter case would suggest that the model will have trouble learning good embeddings for the rare/possibly unseen events during training.

**Limitations:**

The reviewer agrees with the authors that there are no imminent societal impact with the current work.

However, the reviewer would recommend the authors to address potential limits of this work, per the check list. It was marked as yes but no reference section. Mainly, given the potentially large number of event types for real-world problems, it seems like the branching factor of K^D is quite limiting.



**Strengths And Weaknesses:**

Pros:
1) Clarity of writing. The paper is very well-written and easy to follow, which should help with reproducing the results of this work.
2) Originality. Although the idea of leveraging variational inference for casual effect inference is not new [1], the application of it to event prediction is worthy of investigation, in addition to the novelty from the proposed hierarchical structure that can scale up to large number of events and the mixture of posteriors as prior.
3) Evaluation results looks promising, especially the visualization of context probabilities over time.

Cons:
1) Lack of synthetic evaluations. For works with casual inference, it's necessary that the authors should experiment with a synthetic evaluation where ground-truth contexts/confounders are known and are manually manipulated. This allow one to verify exactly how good are the estimated contexts compared to ground-truths and how the model adjust for the context to arrive at the next-step prediction. I don't recall seeing it in the paper.
2) Lack of insights for the real-world datasets in the evaluation section. Is there a way to relate how the changing context causes certain events  to happen more/less? Although the proposed approach did well at detecting shifts and adjusting for these shifts to improve event predictions, can we draw any practical insights from these shifts w.r.t the real-world datasets like movies reviews / preferences.

[1] Louizos, C., Shalit, U., Mooij, J. M., Sontag, D., Zemel, R., & Welling, M. (2017). Causal effect inference with deep latent-variable models. Advances in neural information processing systems, 30

---

> ### Author Response · Authors · 2022-08-02
> **Response to Reviewer t8MQ (Part 1 of 2)**
>
> Thank you for the positive comments on our clear presentation, novel ideas, and solid experiments. In response to your raised questions, weaknesses, and limitations, we provide detailed answers below and supplement new experiments.
>
>
> > ***Q1: “Could the authors comment on the model's performance where the context shifted to a point where previously common events now become rare and previously rare events now become common”***
>
> We answer this question respectively from theoretical and implementation perspectives.
>
> **(Theoretical Perspective)** In the following we will show despite we did not explicitly consider event frequency, our framework is general enough to incorporate and mitigate event frequency shift (i.e., event frequency change due to context shift).
>
> - Fact 1: As is known, there are normally two type of distribution shift [1], namely covariate shift where $P_{train}(S)\neq P_{test}(S), P_{train}(Y|S)= P_{test}(Y|S)$ and concept shift where $P_{train}(Y|S)\neq P_{test}(Y|S), P_{train}(S)= P_{test}(S)$. The problem formulation we consider in section 2 is a mixture of both, since both covariate shift and concept shift can cause $P_{train}(Y, S) \neq P_{test}(Y, S)$.
> - Fact 2: Formally, the problem mentioned by the reviewer (i.e., event frequency shift) can be formulated as $P_{train}(X)\neq P_{test}(X)$, where $X$ denotes random variable of event. It is actually a sufficient condition for covariate shift $P_{train}(S)\neq P_{test}(S)$ in the sense that it is impossible $P_{train}(X)\neq P_{test}(X)$ and $P_{train}(S) = P_{test}(S)$ hold at the same time. It is also intuitively reasonable since rare events in the sequence tend to make the sequence itself rarer.
>
> Therefore, since our framework aims to address temporal distribution shift $P_{train}(Y, S) \neq P_{test}(Y, S)$, theoretically it is also able to address event frequency shift (i.e., “context shifted to a point where previously common events now become rare and previously rare events now become common”).
>
> [1] A unifying view on dataset shift in classification, Pattern Recognition 2012
>
> **(Implementation Perspective)** In terms of implementation (i.e., formulation of learning objective), if we compare the objective with and without causal intervention (i.e., do operation), ours has an additional scaling factor $P(C)/P(C|S)$, meaning the model tends to “up-weight” minority context in training. Since context shift is the cause of the difference in event frequency, it also means the model tends to “up-weight” those samples with rare events, so that all events tend to be equally trained.
>
> >***Q2: "Experiment with a synthetic evaluation where ground-truth contexts/confounders are known and are manually manipulated."***
>
> We conducted an additional experiment based on a synthetic dataset generated by four different models trained on different parts of Yelp dataset for a quick (but not rigorous) verification. We manipulate the temporal distribution shift by treating two of these models as majority for generating training sequences and minority for testing sequences. We then re-train a model on the sythetic dataset with $K=4$. We compare the true and estimated frequency of context (where their correspondences are manually chosen) in the testing set, and the results are as follows, which show our model can to some extent recover the latent context in this particular setting.
>
> |  | Context 1 | Context 2 |Context 3 | Context 4 |
> | -------- | -------- | -------- |-------- | -------- |
> | Ground-Truth     | 0.4     | 0.4     | 0.1 | 0.1 |
> | Estimated     | 0.33     | 0.39     | 0.15 | 0.13|
>
>
> > ***Q3: “Is there a way to relate how the changing context causes certain events to happen more/less?” and “can we draw any practical insights from these shifts w.r.t the real-world datasets like movies reviews / preferences”***
>
> Yes, one can statistically analyze the relation between changing context and event frequency by, e.g., using a certain branch of our model to generate event sequence, and even measure this relation by quantifying the context shift and event frequency shift.
>
> But unfortunately, due to the nature of the problem setting where the context is defined as a latent and abstract factor (which is a major challenge we aim to tackle and serves as our contribution), it would be hard or even impossible to identify its meaning in the real world. Hence, we can hardly attribute event frequency shift to certain meaningful context or other explainability purposes that involve the practical meaning of the context. However, we think it is actually an advantage if the goal is to handle temporal distribution shift since it allows flexibility and capability for exploring latent context and distribution shift in a data-driven manner.

---

> > ### Author Response · Authors · 2022-08-02
> > **Response to Reviewer t8MQ (Part 2 of 2)**
> >
> > > ***Q4: "the reviewer would recommend the authors to address potential limits of this work, per the check list"***
> >
> > For clarity, we list and summarize the limitations and potential societal impacts in the last section of Appendix in our new version.
> >
> > > ***Q5: "Mainly, given the potentially large number of event types for real-world problems, it seems like the branching factor of K^D is quite limiting"***
> >
> > Indeed, this is a limitation of our work and remains a major impediment for all works that are dealing with infinitely many discrete confounders. We also want to remark that ours is already an improvement over existing causal frameworks by increasing $K*D$ to $K^D$. In practice, we also found (in section 4.5) that increasing $K$ and $D$ within a certain range is useful, but will harm the performance if we keep increasing $K^D$ possibly because of the bottleneck of the dataset.

---

> ### Author Response · Authors · 2022-08-06
> **A kind reminder before the discussion phase ends**
>
> Dear reviewer t8MQ,
>
> Thanks again for your review. Since the discussion period is approaching its end, we would be glad to hear from you if we have addressed your questions/concerns.
>
> Kind regards, The Authors

---

### Official Review · Reviewer_XNzH · 2022-07-16

**Rating:** 4
**Confidence:** 4
**Soundness:** 2 fair
**Presentation:** 3 good
**Contribution:** 3 good

**Summary:**

This paper aims to address the task of event sequence modeling, where the aim is to predict the next event in sequential data, focusing on the scenario where there is a substantial gap between the time the data is collected and when the prediction is made.  In this setting, the data distribution may have shifted over time, and the events' context may also have changed.  The authors propose a strategy that combines several ideas: a causal "backdoor adjustment" to correct for the confounding nature of the latent context, variational inference to infer the distribution over the latent context, and a neural network with a "hierarchical branching structure" for the inference.

**Questions:**

Can the authors identify a scenario in which this approach is more appropriate than a continuous-time model that leverages timestamps to model the gap between training and deployment and also models the context's temporal dynamics?

**Limitations:**

Since there is little focus on the potential application of the work, there is not much to say about its societal impacts.


**Strengths And Weaknesses:**

The proposed methodology combines a diverse range of ideas from different subfields of machine learning (temporal modeling, causality, variational inference, neural networks) into a sophisticated, cohesive approach.  The proposed methodology seems reasonable (though with this many moving parts, I could easily have missed something if there was an issue).

While the problem formulation is interesting, I am not convinced that it is well motivated for real applications.  Arguably, if the goal is to predict events in the remote future, a model which predicts the next event in a sequence is not appropriate, since the event to be predicted will most likely not actually be the "next" event.  In this scenario, a continuous time model that uses timestamp information would be more appropriate to properly model how the gap between training and prediction would impact the prediction.  E.g., in the example where the context is the season, one can easily incorporate seasonality into a model that takes the timestamp into account.  There may be specific applications where the proposed method is needed, but the paper has not provided one.

I am also not convinced that the method needs to be this complicated, and the experimental results have not confirmed this.  To justify this complicated formulation, it would have been valuable to include comparisons to simplifications of the proposed framework.

The analysis regarding the limitations of maximum likelihood estimation (Section 2.1) was very nice, though not really surprising.  If you train a model that does not account for latent confounders in a scenario where those confounders have changed, it is clearly not going to work well.  The issue is not really with maximum likelihood estimation, but with the misspecification of the model in not modeling the confounder, as well as using a non-causal method in a scenario where causality is needed.

The paper is reasonably well written.  If it is hard to follow in parts, this is mostly because of the complicated combination of disparate ideas.

Strengths:

  - Innovative approach combines ideas from different subfields in a unique way.

  - The use of causal methods for this problem is worth pursuing.

  - Experimental results include both quantitative and qualitative analyses, and investigate scalability and hyper-parameter settings.

Weaknesses:

  - The problem formulation is not well motivated (see above).  It is a lot of effort to try to fix a model which is not appropriate for its task (event sequence modeling for predicting events in the remote future) instead of just using an appropriate model (continuous time models, particularly those which explicitly model the context and distribution shift).  Those types of more traditional models should also be compared to as baselines.

  - The approach, although interesting, is complicated, and there are no experimental results that help to justify this complexity.

---

> ### Author Response · Authors · 2022-08-02
> **Response to Reviewer XNzH (Part 1 of 3)**
>
> Thank you for the time, thorough review, and constructive suggestions. Also, thanks for appreciating our novel methods, significance, and extensive experiments. We noticed that your main concerns lie in our motivation for the problem formulation and more experiments for verifying the necessity of proposed components. In the response below, we first provide a detailed and organized clarification for our problem motivation (including all related questions and weaknesses), then we supplement new ablation study results. We hope these answers could be valuable for your re-assessment of our work.
>
> ### 1. Clarification on Motivation and Beyond
>
> > ***Comment 1. "I am not convinced that it is well motivated for real APPLICATIONS." and "There may be specific APPLICATIONS where the proposed method is needed, but the paper has not provided one."***
>
>
>
> (***Target application scenario***)
> Predicting the next type of event based on historical event sequence (a.k.a. event prediction) is actually a well-established problem with a variety of applications (e.g., epidemic control, clinical treatment). Perhaps the most prominent application under active research is (sequential) recommender system [1-4], where **the task is EXACTLY to use a fixed length of most recent interaction sequence (i.e., event sequence) to predict a new item the user is likely to interact (i.e., the next event)** for recommendation. Crucially, in these applications and recommendation datatsets used in our experiment, **the timestamp information is in default NOT available** or converted into a form that is easier to handle (e.g., season, age) as part of event feature.
>
>
> [1] Sequential Recommender Systems: Challenges, Progress and Prospects
> [2] Self-Attentive Sequential Recommendation, ICDM'18
> [3] BERT4Rec: Sequential Recommendation with Bidirectional Encoder Representations from Transformer, CIKM'19
> [4] Deep Interest Evolution Network for Click-Through Rate Prediction, AAAI'19
>
>
>
> > ***Comment 2. “if the goal is to predict events in the REMOTE FUTURE, ... , the event to be predicted will most likely not actually be the next event" and “fix a model which is not appropriate for its task ... for predicting events in the REMOTE FUTURE”***
>
> (***Clarifying the definition of “remote future”***)
> Based on the above comments, we suspect there might be a misunderstanding regarding the definition of “remote future” and our problem setting. For problem setting, in this work, **predicting the next event is the “task”** and **temporal distribution shift is the “challenge”** (which will be discussed in detail next), NOT "trying to use next event prediction for the task of solving temporal distribution shift". The next clarify the definition of "remote future":
>
>
> [Definition A (a different problem)]: The time-stamp of $\hat y$ is significantly greater the time-stamp of last event $x_t$ in the input sequence $S$. Example: using a user’s behavior in 2020 as input $S$ to predict his/her behavior $\hat y$ in 2022.
>
> [Definition B (ours)]: The time-stamp of testing data $(S_{test},\hat y)$ is significantly greater the time-stamp of $(S_{train}, y_{train})$. Example: training the model by using data collected in 2020, and using user's behavior in 2022 (i.e., $S_{test}$) to predict next behavior in 2022 (i.e., $\hat y$).
>
> We focus on definition B that is essentially different with definition A in the sense that **it does NOT make any assumption on the time gap between last event and prediction**. This is indeed a widely-recognized challenge ([41, 36, 22] in the main paper) in both research and industrial fields reflected by the widely observed discrepancy between offline and online performance, which unfortunately is neither well-understood nor properly solved.
>
> We also have given a formal formulation of definition B in appendix G with a new illustrative figure.9 for readers to better understand the problem formulation.

---

> > ### Author Response · Authors · 2022-08-02
> > **Response to Reviewer XNzH (Part 2 of 3)**
> >
> > > ***Comment 3. "... a continuous time model that uses timestamp information would be more appropriate to properly model how the gap between training and prediction ..." and "... If you train a model that does not account for latent confounders in a scenario where those confounders have changed, it is clearly not going to work well ..."***
> >
> > (***The role of timestamp and “explicit context” information***)
> > We appreciate the reviewer hinting on that explicitly considering time-stamp information (e.g., explicitly incorporate seasonality) may help to mitigate the distribution shift. More generally, there might exist other explicit contexts that can not be directly extracted from timestamp but is helpful for mitigating the shift such as user’s preference, fashion trend, etc.
> >
> > But the key question is **"what if we don't have any such information about time-stamp or other explicit contexts, or such context is abstract and infeasible to describe?".** This is the major challenge we actually face in event prediction (no explicit context including time-stamp), and is tackled by our variational context adjustment approach. We also remark that our method **can be combined with existing methods (e.g., some continuous-time models) that consider explicit contexts including time-stamp**, but this is beyond the scope of this work.
> >
> >
> > > ***Question. "Can the authors identify a scenario in which this approach is more appropriate than a continuous-time model that leverages timestamps to model the gap between training and deployment and also models the context's temporal dynamics?"***
> >
> > To conclude, we can't say our model is more or less appropriate than a continuous-time model since they are orthogonal used in different settings and tackling different technical challenges:
> > 1. Our model is used for “(next) event prediction” task where time-stamp is not necessarily available, while continuous-time model assumes its availability.
> > 2. Our model aims to deal with "distribution shift" in definition B with no assumption on the time gap between sequence and prediction, and continuous-time model seems to a reasonable choice in definition A.
> > 3. Our model tackles the challenge of implicit context, and continuous-time model deals with explicit context that is related with time-stamp.

---

> > > ### Author Response · Authors · 2022-08-02
> > > **Response to Reviewer XNzH (Part 3 of 3)**
> > >
> > > ### 2. Ablation Study and More Discussions
> > >
> > > > ***Comment 4. "To justify this complicated formulation, it would have been valuable to include comparisons to simplifications of the proposed framework."***
> > >
> > > **Ablation studies in the main paper. (Ablation on Structure)**
> > >
> > > In fact, we have already conducted some ablation studies (though not explicitly pointed out) in the experiment section to justify modules of the framework. We summarize them as follows:
> > >
> > > 1. In section 4.1, the backbone models we use for recommendation task are indeed simplifications of our CaseQ by **removing branching modules in the architecture** (the KL-divergence term in loss function is naturally inapplicable). We can see from table 1 there is a remarkable decrease in the percentage of performance drop as we enlarge the time gap between training and testing data, compared with these backbone models.
> > > 2. In section 4.2, baselines DMoE and multi-head Transformer could be considered as simplifications of our CaseQ **in terms of branching unit design** (since they have the same inference unit, width, and depth). The difference is that they are not able to dynamically learn context. Our method is still more robust as shown in figure 3.
> > > 3. In section 4.4, when $K=1$, our model degrades to **a basic model where there is only one context**, and as we can see from figure 7, the performance drop of this basic model is more significant compared with models with larger $K$. The effectiveness of our model with increasing depth $D$ also justifies our hierarchical branching structure design.
> > >
> > > **More results. (Ablation on Loss Function)**
> > >
> > > To see the effect of the additional KL-divergence term in the loss function in Eq.(16), we conduct more experiments to see the performance variation of CaseQ. Results are shown in the appendix of our updated version with a new plot. We found that our approach is less robust when removing this term ($\alpha=0$). Moreover, by gradually increasing $\alpha$ in a certain range, our approach becomes more robust w.r.t. distribution shift (i.e., less performance drop with larger gap size). This shows **the KL-divergence term in the objective (that is derived from the causal framework and integrated with our structure) is crucial** for the effectiveness of handling distribution shift.
> > >
> > > **More discussion on the complexity of model**
> > >
> > > Though our model may seem complex in appearance, it introduces marginal extra parameters compared with similar-structured base models (i.e., Multi-head Transformer, Ensemble, DMoE), and could also be efficiently **trained in an end-to-end manner**. In our experiments, we control the **complexity of our model to be the same as baselines’** to ensure fair comparison (see section 4.2). Also, we conduct scalability test in section 4.4 and found that both **training and inference time scale at a sub-linear rate with more contexts**. While our training time cost is indeed larger than same-sized baselines because of different formulation of the loss function, **the inference time is very close** while ours is considerably more robust to distribution shift.

---

> ### Author Response · Authors · 2022-08-06
> **A kind reminder before the discussion phase ends**
>
> Dear reviewer XNzH,
>
> Thanks again for your review. We have provided informative responses to clarify your potential misunderstanding of our motivation for the problem formulation. We have also summarized ablation studies with newly added experimental results for justification. Since the discussion period is approaching its end, we would be glad to hear from you if we have addressed your concerns.
>
> Kind regards,
> The Authors

---

### Author Response · Authors · 2022-08-02
**Message to area chairs and all the reviewers**

Dear Area Chairs and Reviewers,

We thank the reviewers for their time, valuable comments, and constructive feedback. We are encouraged to see that generally, the reviewers found our paper well written (XNzH, t8MQ, HB3W), the problem significant (XNzH, k7dp), our models reasonable (XNzH, k7dp) and novel (XNzH, t8MQ, HB3W, k7dp), the evaluation solid and extensive (XNzH, k7dp), and the results promising (t8MQ, HB3W, k7dp).

In light of the nice suggestions from the reviewers, we modify our paper accordingly with a newly uploaded version. In the response below, we address all the questions point-by-point and add extra experiment results. We also welcome further comments or requests.

---

### Author Response · Authors · 2022-08-08
**New Draft Uploaded**

Dear Reviewers,

Thanks again for your detailed review and constructive suggestions for improvement. A new version of our paper has been uploaded with modified parts colored blue, based on our rebuttal. We will also further refine these parts and implement your suggestions in the final version.

Sincerely,
The Authors

---

### Meta-Review · Area_Chair_zjja · 2022-08-25

**Recommendation:** Accept
**Confidence:** Certain

**Metareview:**

This paper proposes a method for predicting the next event given sequential data. For the prediction under distribution shift, the proposed method uses a backdoor adjustment, variational inference of latent context, and hierarchical branching structure. The proposed method that combines techniques from different fields is interesting. In particular, the use of causal methods for the distribution shift problem in temporal event prediction is novel. The experimental results demonstrate the effectiveness of the proposed method well quantitatively and qualitatively. The paper should be improved according to the reviewers’ comments, e.g., clarifying the motivation in real-world applications.

**Award:**

No

---

### Decision · Program_Chairs · 2022-09-14

Accept